# A Detailed Study of Rainbow Trout (*Onchorhynchus mykiss*) Intestine Revealed That Digestive and Absorptive Functions Are Not Linearly Distributed along Its Length

**DOI:** 10.3390/ani10040745

**Published:** 2020-04-24

**Authors:** Nicole Verdile, Rolando Pasquariello, Marco Scolari, Giulia Scirè, Tiziana A. L. Brevini, Fulvio Gandolfi

**Affiliations:** 1Department of Agricultural and Environmental Sciences, University of Milan, 20133 Milano, Italy; Rolando.pasquariello@unimi.it; 2Skretting Aquaculture Research Centre, 37100 Verona, Italy; Marco.Scolari@nutreco.com (M.S.); giulia.scire@skretting.com (G.S.); 3Department of Health, Animal Science and Food Safety, University of Milan, 20133 Milano, Italy; tiziana.brevini@unimi.it

**Keywords:** intestine, epithelium, folds, renewal, rainbow trout

## Abstract

**Simple Summary:**

Aquaculture is the fastest growing food-producing sector due to the increase of fish intended for human consumption. However, aquaculture growth generates concerns, since carnivorous fish are extensively fed using fish-meal and fish-oil. This constitutes a severe limit to the aquaculture industry, questioning its sustainability. Consequently, alternative feeds are continuously searched through extensive in vivo feeding trials. Undoubtedly, to evaluate their impact on the gastrointestinal tract health, detailed knowledge of the intestine morphology and physiology is required. To date, extensive studies have been performed in several livestock species; however, available information on fish is limited nowadays, most importantly because their alimentary canal is able to easily adapt to external stimuli, and their intestinal morphology is affected by external factors. Therefore, it is essential to establish accurate reference values, especially along the productive cycle of animals raised in standardized conditions. Here, we performed a detailed characterization of the epithelial cells lining the intestinal mucosa in rainbow trout along the first year of development. We studied the absorptive and secretory activity as well as its ability to self-renewal. Our results indicate that, in this species, both digestive and absorptive functions are not linearly distributed along the intestinal length.

**Abstract:**

To increase the sustainability of trout farming, the industry requires alternatives to fish-based meals that do not compromise animal health and growth performances. To develop new feeds, detailed knowledge of intestinal morphology and physiology is required. We performed histological, histochemical, immunohistochemical and morphometric analysis at typical time points of in vivo feeding trials (50, 150 and 500 g). Only minor changes occurred during growth whereas differences characterized two compartments, not linearly distributed along the intestine. The first included the pyloric caeca, the basal part of the complex folds and the villi of the distal intestine. This was characterized by a significantly smaller number of goblet cells with smaller mucus vacuoles, higher proliferation and higher apoptotic rate but a smaller extension of fully differentiated epithelial cells and by the presence of numerous pinocytotic vacuolization. The second compartment was formed by the proximal intestine and the apical part of the posterior intestine complex folds. Here we observed more abundant goblet cells with bigger vacuoles, low proliferation rate, few round apoptotic cells, a more extended area of fully differentiated cells and no pinocytotic vacuoles. Our results suggest that rainbow trout intestine is physiologically arranged to mingle digestive and absorptive functions along its length.

## 1. Introduction

Rainbow trout (*Oncorhynchus mykiss*) is one of the most widespread species in aquaculture due to its many merits that include adaptability to the farming environment, reproductive efficiency and disease resistance [1]. Nevertheless, the pressure to optimize the farming efficiency is constant and involves different aspects, including the continuous refinement of the diet [2]. Environmental sustainability and costs are the main drivers in the search for the optimal aquafeed, while progress depends on the accurate knowledge of intestine physiology in this and other related farmed species.

The gastrointestinal tract (GI) promotes the absorption of nutrients acting as a selective filter between the lumen and the circulatory system, but it simultaneously avoids the passage of harmful intraluminal xenobiotics [3]. Extensive and accurate studies on intestinal morphology have been performed in different livestock species [4], such as poultry, ruminants and pig [5], whereas in fish, a detailed morphological and functional characterization of the intestinal wall is still limited. In fact, fish being the largest group of vertebrates [6], are an extremely numerous and heterogeneous group. Several studies focused on the Teleostei intestinal features showed a wide variability among species due to diet, phylogeny and body shape [6]. Their GI can quickly and reversibly adapt to the environmental changes based on their physiological requirements [7]. The morphology of their alimentary canal is determined by different factors, such as taxonomy, feeding habitats, food type and frequency of food intake [8], making its detailed study and characterization complicated.

Current histological parameters used to determine the rainbow trout gut health in correspondence of dietary modifications are based on classical morphological changes. These include villus shortening, widening of lamina propria of villi, nuclear position disparity, intraepithelial lymphocyte infiltration, qualitative changes in mucus secretion [9] and massive enterocyte vacuolization. Parameters are accurately measured in the proximal intestine, whereas the analysis of the distal intestine is less detailed, possibly due to its complex morphology. This distal intestine is the major site of macromolecules absorption in most fish species, including salmonids [7]. In addition, only a few studies have been performed on rainbow trout intestine report quantitative data [10], while most of them are based on qualitative observations [6]. Finally, these studies are rarely carried out in animals raised in optimal and standardized conditions and do not consider the relationship between intestinal morphology and fish growth. Therefore, the aim of this work was to characterize the intestinal epithelial cells in rainbow trout to establish accurate reference values and to identify novel markers for increasing the sensitivity of earlier investigations. To this purpose, we studied individuals raised in standard conditions and ranging from 50 to 500 g in weight, the size range used for feeding trials.

## 2. Materials and Methods

### 2.1. Samples Collection

Five female rainbow trout (*Oncorhynchus mykiss*) for each age (7, 10, 12 months, weighing approximately 50, 150 and 500 g respectively) were euthanized under total anesthesia according to Annex IV EU guideline 2010/63, during non-experimental clinical veterinary practices. Fish were maintained in 600 L volume tanks with a water flow of 700 L/h and were reared under a photoperiod regimen of 24 h light. The water temperature was maintained constant at 15 °C.

These stages correspond to time points used in in vivo feeding trials. Moreover, in our experiment, these steps correspond to stepwise increase of digestible energy: 18 MJ/kg up to 50 g, 18.5 MJ/kg from 50 to 150 g and 19 MJ/kg up to 500 g (Optiline, Skretting, Verona, Italy). Animals included in this study were healthy and were raised in optimal conditions at Skretting Italia Spa (Verona, Italy). Immediately after sacrifice, a longitudinal incision along the ventral line was performed and the whole gastrointestinal (GI) tract was removed.

At sampling time, each animal included in this study was weighed and its length measured. Once the whole intestinal tract was removed, the length of each tract was also defined. All the observed values are reported in Table 1 and are expressed as mean ± SD.

We collected segments of 2 cm of both anterior and posterior intestine. We defined as anterior intestine the tract comprised between the pyloric sphincter and the ileum-rectal valve. For consistency, we sampled only the segments with annexed pyloric caeca. We considered as distal intestine the tract downstream of the ileum-rectal valve characterized by larger diameter, darker pigmentation and circularly arranged blood vessels. For consistency, we collected all samples in the middle of this tract (Figure 1). Samples were immediately fixed in 10% neutral buffer formalin for 24 h at room temperature, dehydrated in graded alcohols, cleared with xylene and embedded in paraffin.

### 2.2. Histology and Histochemistry

After dewaxing and re-hydration, 5 µm thick sections were stained with hematoxylin/eosin (HE) to evaluate the morphological features. Other sections were stained with Periodic acid–Schiff (PAS) or with Alcian Blue pH 2.5 (AB) for histochemical purposes. Subsequently, new slides, were stained with Periodic acid–Schiff (PAS)/Alcian Blue pH 2.5 staining kit (Bio Optica, 04-163802) to analyze the overall complex carbohydrates [11,12].

Lendrum’s staining was used to identify acidophilic granules containing cells. Sections were incubated with Phloxine B to visualize acidophilic compounds and then with Tartrazine to remove the non-specific staining.

The alkaline phosphatase expression was used as a marker for the identification of fully differentiated epithelium. Briefly, slides were rehydrated and brought to distilled water, were then immersed in fresh Tris HCL (pH 9.5) solution to create the alkaline environment for 5 min. They were then incubated with BCIP/NBT substrate (5-bromo-4-chloro-3-indolyl phosphate/nitroblue tetrazolium, (Vector Laboratories, SK-4500 USA), which produces an indigo reaction product in the presence of alkaline phosphatase (AP) enzyme. Sections were then rinsed in tap water, counterstained using Mayer’s hematoxylin, dehydrated and mounted.

### 2.3. Morphometric Evaluation

Images were acquired using a NanoZoomer S60 Digital slide scanner (Hamamatsu photonics, Hamamatsu city, Japan)) and observed at continuous magnifications between 20× and 80×. Villi morphometric evaluation was performed on HE-stained sections using the NDP.view software (Hamamatsu, Japan). For each fish, we measured 20 villi for each intestinal tract and 10 complex folds in the distal intestine. Villus height was calculated from the apex of the villus to the villus-intestinal folds junction; villus width was measured at the base, at the middle and at the apex and it was expressed as the mean of the three measurements. Fold height was measured from the compactum layer of the submucosa to the apex of the folds. Goblet cells were counted along 1 mm of epithelium for each intestinal compartment in each fish. The precise length of the epithelial lining was measured using the specific function of the NDP.view software. Volume of goblet cell mucus vacuoles was estimated using the same software. For each sample, 50 goblet cells were evaluated in all intestinal sections considered.

### 2.4. Quantitative Stereological Analysis

Systematic sampling was performed as described in detail previously [13]. Briefly, the estimation of the volume densities (Vv) of the separate layers of the intestine was based on the general principle of Delesse [14]. The volume of each layer was estimated from the fractional area of the structure of interest (e.g., the mucosa) and the total area of the reference compartment (e.g., the intestinal wall) was measured in histological cross-sections. Area-measurements were performed by point-counting [13], as specified below. For estimation of the volume densities of the different layers, the regions of interest containing the respective reference compartments were defined in the histological sections. Within these regions of interest, systematically randomly sampled areas (70% of the total sectional area of the region of interest) were photographed and superimposed with an adequately sized grid of equally distant points. The number of points hitting the interested structure and the respective reference compartment were counted and the fractional area of the structure of interest and the total area of the reference compartment were then calculated from the respective quotient of points hitting these structures. The magnification was chosen to allow the relevant portion of the intestinal wall to be contained in each field of vision. Vv were expressed as percentages and were calculated as follows:Vv_(analyzed compartment, reference compartment)_ = [∑P_(analyzed compartment)_/∑P_(reference compartment)_] **×** 100
where ∑P_(analyzed compartment)_ is the number of points hitting the compartment under study, and ∑P_(reference compartment)_ is the number of points hitting the relevant structure.

### 2.5. Immunohistochemistry

Proliferating cell nuclear antigen (PCNA) localization was characterized by indirect immunohistochemistry using the Avidin Biotin Complex method (VECTASTAIN^®^ Elite^®^ ABC, Vector Laboratories, Burlingame, CA, USA) following manufacturer instructions. Briefly, slides were brought to boiling in 10 mM sodium citrate buffer, 0.05% Tween20 (pH 6) in a pressure cooker for 1 min for antigen retrieval. After cooling at room temperature for 20 min, sections were rinsed in PBS (phosphate-buffered saline, pH 7.4) and then were immersed in 3% H_2_O_2_ solution in methanol for 15 min to quench the endogenous peroxidase. To prevent aspecific binding, sections were then incubated in Normal Blocking Serum (Vectastain ABC Elite KIT, Burlingame, CA, USA) at room temperature for 30 min. Sections were incubated with Anti PCNA Mouse monoclonal antibody 1:1600 (Millipore Corporation, MAB424, Darmstadt, Germany) diluted in 4% BSA in PBS with 0.05% Tween20, for 60 min at room temperature in a humid chamber. Sections were then incubated with appropriate biotinylated secondary antibody for 30 min at room temperature in a humidified chamber followed by the avidin-biotinylated horseradish peroxidase (HRP) complex (Vectastain ABC Elite KIT, Burlingame, CA, USA) for another 30 min. Finally, sections were incubated with ImmPACT NovaRED substrate (Vector Laboratories, SK-4105, Burlingame, CA, USA), which produces a red reaction product in the presence of peroxidase (HRP) enzyme. Sections were then briefly counterstained with Mayer’s hematoxylin, dehydrated and permanently mounted with a mounting media (Bio-Optica, 05-BMHM100, Milano, Italy). Secondary antibodies controls were performed following the same staining protocol but omitting the primary antibody.

### 2.6. TUNEL Test

Peroxidase in situ detection Apoptosis kit (Millipore Corporation, S7100, Darmstadt, Germany) was used to detect cells undergoing apoptosis following the manufacturer instructions. In brief, sections were brought to boiling in 10 mM sodium citrate buffer, 0.05% Tween20 (pH 6) in a pressure cooker for 1 min to improve the access of the terminal deoxynucleotidyl transferase (TdT) to the fragmented DNA. Slides were then cooled at room temperature for 20 min, washed in PBS, and incubated in a humidified chamber for 15 min in 3% H_2_O_2_ in methanol to quench endogenous peroxidase. Sections were rinsed in distilled water, exposed to equilibration buffer (Millipore Corporation, S7100, Darmstadt, Germany) for 20 s and incubated with TdT enzyme digoxigenin-conjugated in a humidified chamber at 37 °C for 60 min. TdT enzyme was previously diluted in reaction buffer (1:32) (Millipore Corporation, S7100, Darmstadt, Germany). Reaction was stopped by immersing sections in fresh prepared stop/wash buffer in a Coplin jar for 10 min. Anti-Digoxigenin peroxidase conjugated antibody was applied to slides for 30 min at room temperature. Samples were then washed in PBS and were incubated with 3,3′-diaminobenzidine solution (ImmPACT^®^ DAB, SK-4105 Vector Laboratoris, Burlingame, CA, USA), which, in the presence of a peroxidase (HRP) enzyme, produces a brown reaction product. Sections were then briefly counterstained with Mayer’s hematoxylin, dehydrated and permanently mounted with a mounting media (Bio-Optica, 05-BMHM100, Milano, Italy).

### 2.7. Statistical Analysis

Quantitative data were expressed as mean ± SD. Results were analyzed by using One-way or Two-way analysis of variance (ANOVA) followed by all-pairwise multiple comparison test with the Holm–Sidak method. The two independent variables considered (animal weight and intestinal tract) were independent of each other, and there was no relationship between the observation in each group or between the groups themselves. Dependent variables considered (mucosa volume, goblet cells number and vacuoles volume and goblet cells producing acid, neutral and a mixed combination of them) were normally distributed according to the Shapiro–Wilk normality test. The homogeneity of variance was assessed through Bartlett’s test. Differences were considered statistically significant if *p* < 0.05.

## 3. Results

### 3.1. Gross Anatomy

Macroscopically, the rainbow trout intestine corresponds to the general description of this organ in teleost fish [15]. It comprised a proximal intestine with blind diverticula called pyloric caeca annexed to its upper part and a distal intestine [15]. The latter is characterized by a larger diameter, dark pigmentation and circularly arranged blood vessels in agreement with a previous study performed in Brown trout [16]. Circular folds protruding from the distal intestinal wall towards the lumen were also evident even if this is not a typical teleost feature.

### 3.2. Microscopical Anatomy

Pyloric caeca, proximal and distal intestine are lined by a tunica mucosa constituted by epithelium and lamina propria forming villi along all tracts.

Villus length in pyloric caeca increased significantly in parallel with age (Table 2). Interestingly, in this region, at 500 gr we observed enterocytes supranuclear vacuolization (Figure 2).

In the proximal intestine, we observed a wide variation of villus length. In order to reduce the wide standard deviation and making possible a meaningful statistical analysis, we divided them into two arbitrary groups: shorter and longer of 400 µm. Average short villi (below 400 µm) length remained constant during growth, whereas long villi (above 400 µm) increased their length significantly when animals reached the 500 gr size (Table 3). At the same time, villi in the larger animals became more branched (Figure 3) whereas short villi were rarer. No supranuclear vacuoles were observed in the proximal intestine enterocytes.

The large circular folds observed macroscopically in the distal tract, corresponded to complex folds formed by tall primary extroflexions of the muscular and mucosal layers. From their main axis, secondary extroflexions of epithelium and lamina propria formed villi like those present along the rest of the intestinal wall. Along the entire length of the posterior intestine, the complex folds were interspersed at regular intervals among normal villi. Also, in the distal intestine, villi were very heterogenous in length, so we divided them into shorter or longer than 400 µm. Their length, as well as the height of the complex folds, remained constant along development. Villi and folds in the distal tract had to be measured on longitudinally embedded samples to appreciate and to measure the typical folds structure. Unfortunately, due to the very small lumen of the 50 g samples, the opening procedure damaged the mucosa morphology to the point of compromising an accurate morphometric analysis of the height of the folds. Therefore, we decided to exclude this data from the analysis (Table 4).

In the distal intestine, villi protruding from the organ wall and those located at the basal part of the complex folds presented pinocytotic vacuoles, whereas those located at the folds apex did not. The latter, on the contrary, were characterized by numerous actively secreting goblet cells as observed in the villi of the proximal intestine (Figure 4).

Along all intestinal tracts, the tunica submucosa was composed of a thick compactum layer interposed between two thin granulosum layers, so-called because of the presence of the characteristic granular cells. The upper granulosum layer was always thinner than the bottom one. Notably, the tunica submucosa present in the complex folds axis was homogeneous with neither compactum nor granulosum layers.

The whole intestinal tract possessed a thick muscle layer composed of an inner circular and outer longitudinal layer. Circularly-arranged muscle cells were also clearly visible within the apical part of the complex folds of the distal intestine but not at their base.

No inflammation features were observed, such as villi shortening and nuclear positioning disparity. Only a few intraepithelial lymphocytes were found confirming the animals’ healthy state.

### 3.3. Quantitative Sterological Analysis

Volume estimation of each intestinal layer indicated that the mucosa was the predominant layer in both intestinal tracts. Moreover, the mucosa volume was significantly higher in the distal than in the proximal intestine at all ages and remained unchanged during development (Figure 5). The bigger mucosa volume of the posterior tract was not due to an alteration of the epithelium:lamina propria ratio since it remained constant in the different groups and indicated a healthy intestinal tract.

### 3.4. Goblet Cells

Goblet cell numbers and vacuole volume were not homogenously distributed among the different intestinal tracts. Goblet cells were significantly more abundant and their vacuoles significantly bigger in the proximal intestine and in the apical part of the posterior intestine complex folds. Villi of the distal intestine and of the basal part of the complex folds shared with the pyloric caeca a smaller goblet cell number and vacuole volume (Figure 6).

The accurate quantitative analysis of goblet cells was limited to the proximal intestine and the pyloric caeca because the number of PAS-positive goblet cells in the distal part was confounded by the presence of numerous PAS-positive pinocytotic cells (Figure 7).

PAS staining revealed that most goblet cells had a low affinity for Periodic acid-Shiff (Figure 8), but most of them strongly reacted with Alcian Blue pH 2.5 (Figure 9). Therefore, in order to have a more detailed qualitative analysis, we performed a combined PAS-Alcian Blue staining. This revealed a range of blue stain intensity in all investigated tracts indicating a heterogenous production of complex carbohydrates. A few cells produced only neutral mucous (PAS positive-magenta cells), some produced acid mucous (AB positive-light blue cells), while the majority secreted a mixed combination of acid and neutral mucous (PAS-AB positive-dark blue cells) (Figure 10).

In the proximal intestine, the distribution of the different types of goblet cell was the same in short and long villi. In both types of villi, however, the number of goblet cells producing acid mucus significantly increased in 500 g individuals in parallel with the onset of villi branching (Figure 11). We did not observe the same trend in the pyloric caeca where the quality of mucus secretion remained constant along development (Figure 12).

### 3.5. Intestinal Eosinophilic Granule Cells

Round cells positive to the Phloxine tartrazine staining were observed in the upper and lower granulosum layers of the submucosa along the whole intestinal length. At high magnification, these cells showed the presence of cytoplasmatic granules enabling their classification as eosinophilic granule cells (EGCs). Their shape and position do not correspond to the typical Paneth cell. These cells, however, were negative for both alkaline phosphatase and peroxidase, suggesting that EGCs do not possess the morphological characteristics of Mast cells (Figure 13).

### 3.6. Proliferation, Differentiation and Apoptosis

A strong PCNA signal was detected within the intestinal folds at the basis of the villi, confirming that the stem/progenitor cell zone is located here at all ages, in all investigated regions. In the proximal intestine, PCNA was not limited to the folds but was also expressed in correspondence of villi branching points. Mature, fully differentiated epithelial cells were identified by the histochemical detection of the alkaline phosphatase (AP) activity. As expected, the signal was localized along the villi length and at their apex.

As for the goblet cells also, the AP/PCNA staining pattern was not homogeneous along the intestine and we could identify two patterns. One characterized by a restricted proliferating area accompanied by an expanded differentiated zone present in the proximal intestine and in the apical region of the complex folds of the distal intestine. The other characterized by an extended proliferating area accompanied by a restricted differentiated zone present in pyloric caeca and distal intestine villi and in the basal part of the complex folds (Figure 14 and Figure 15).

Cells undergoing apoptosis were identified using the TUNEL assay and showed different patterns along the intestine. Only a few cells undergoing apoptosis were visible in the proximal intestine and on the apical part of the complex folds, whereas, many more were detected in the pyloric caeca, in the distal intestine villi and on the basal part of the folds (Figure 16 and Figure 17). Apoptotic cells presented two different morphology: round and slender. Round cells were present along the entire intestinal mucosa and could be detected exfoliating into the intestinal lumen (Figure 18). Slender cells were found exclusively in the pyloric caeca, in the villi and in the basal part of the complex folds of the distal intestine. The same cells displayed a dark eosinophilic cytoplasm if stained with eosin, periodic acid-Schiff or phloxine.

Overall, proliferation, differentiation and apoptotic rate were not homogenously distributed along the intestinal epithelium. The pyloric caeca, the distal intestine wall as well as the basal part of the complex folds were characterized by high proliferation and high apoptotic rate but low alkaline phosphatase expression, which also appeared to be fragmented. In contrast, the proximal intestine and the apical part of the complex folds were characterized by a low PCNA expression, few round apoptotic cells and a strong alkaline phosphatase signal (Figure 19).

## 4. Discussion

Here, we performed a detailed characterization of the intestinal epithelial cells lining the intestinal tract in rainbow trout along the first year of development.

Macroscopically, the morphology of rainbow trout intestine corresponds to the general description of this organ in teleost fish [17] with the peculiar presence of complex folds protruding from the distal intestinal wall towards the lumen. These structures have been previously described in close species like *Salmo trutta and Salmo salar* [9,16] as well as in more distant ones like sharks [18] but are not common to all teleost species [17,19,20]. At present, their detailed morphology remains poorly known, and their function is even less understood [16]. It is interesting to note that a circular tunica muscularis was present at the apex of these folds, while the tunica submucosa was reduced to a thin layer of connective tissue with neither the compactum nor granulosum layers. This arrangement suggests that the complex folds can contract independently from the rest of the distal intestine wall, possibly separating two functional compartments [16].

The wide variation of villi length observed in the proximal intestine prompted us to divide the villi into two populations based on their length. This enabled us to notice that, at 500 gr, only the long villi increased their average length, while the number of short villi diminished. At the same time, villi fusion and branching became evident. Some authors sustain that villi fusion represents a typical inflammation sign [9,21], whereas others support the hypothesis that mucosa folds simply get more complex in parallel with growth [16]. Based on our observations, we hypothesize that branching may be accompanied by or due to the short villi fusing into the long ones suggesting possible functional differences between the two villi types. Consistent with previous observations in Salmo salar [15], we did not observe villus branching at any age in the pyloric caeca, but in this tract, supranuclear pinocytotic vacuolization became visible in enterocytes of 500 gr individuals that could be related to lipid accumulation [9]. Similar vacuolization was present in the enterocytes lining the villi and the basal part of the complex folds of the distal intestine, but not in the proximal intestine nor in the apical part of the complex folds. In the adult goldfish, Carassius auratus these vesicles are known to be specialized in the uptake of large intact molecules [22]

Goblet cells followed two different patterns: numerous, swollen, actively secreting cells, in the proximal intestine and in the apical part of the distal intestine complex folds; scarce and inactive in the pyloric caeca and in the rest of the distal intestine. Our observation that the number of goblet cells was significantly more abundant in the proximal than in the distal intestine is in contrast with other data present in the literature [8,19]. However, a recent study suggests that the greater abundance of goblet cells within the posterior tract could be a misinterpretation due to the massive presence of pinocytotic vacuoles in the apical part of the enterocyte’s cytoplasm [23]. Moreover, goblet cell density is strongly affected by several external factors comprising nutrition, physiology and immunology [24,25], and this must be considered as a possible source of variability. On the contrary, our finding of goblet cells producing different mucous substances agrees with previous results in several teleost fish, including rainbow trout [4,8,10]. Interestingly, the significant increase of acid glyconjugates observed in the anterior intestine, the increase of villus branching and the appearance of vacuolized enterocytes in the pyloric caeca all occurred in correspondence with the increase of lipid concentration in the diet. Consequently, since qualitative changes in mucins composition are associated with gastrointestinal disorders, our data suggest the hypothesis that the last diet change may have caused a mild stress [24].

We identified eosinophilic granule cells (EGCs) within the submucosa granular layers. in agreement with previous observations in the *Salmo salar* [26]. EGC are considered the functional equivalent of Paneth cells and indeed, EGCs found in Rainbow trout were positive to Phloxine-tartrazine staining that is specific for Paneth cells. They play a pivotal role in intestinal mucosa defense by secreting antimicrobic peptides and in intestinal stem cells regulation [27]. However, their position outside the epithelium is not compatible with that of *bona fidae* Paneth cells. Recent data report that the stratum granulosum hosts numerous eosinophilic mast cells involved in the inflammatory process [28], so that some authors simply consider eosinophilic granule cells to be mast cells [29]. Our data do not support this conclusion because rainbow trout ECG did not contain alkaline phosphatase nor peroxidase as opposed to mast cells described in fish.

The localization of proliferative cells at the base of the villi in the pyloric caeca and in both anterior and posterior intestine confirmed that the stem/progenitor cells zone is located within the intestinal folds in all ages, in agreement with previous reports in *Salmo salar* [16] and *Salmo trutta* [30]. However, cell proliferation was not homogeneous in the different intestinal districts but followed the dual pattern described for goblet cells: in the proximal intestine and apical part of the distal intestine complex folds, proliferation was restricted to the bottom of intestinal folds, whereas in the pyloric caeca, in the rest of the distal intestinal the signal was more spread along the villus length. Interestingly, spots of cell proliferation were observed also along the villi length but only in the proximal intestine of 500 gr individuals. We hypothesize that these spots correspond to newly formed branches.

Alkaline Phosphatase (AP) is considered a marker for mature enterocytes [31]; however, we observed a clear signal on the apical part of all differentiated cells. This is in agreement with previous observations in mouse small intestine [32]. As expected, the extension of the cell proliferating compartment was inversely proportional to that of fully differentiated cells. However, it was interesting to note that, once again, this distribution also followed a double pattern as described for the other aspects: the anterior intestine and the apical part of the distal intestine complex folds showing low proliferation and extensive differentiation while the opposite occurred in the pyloric caeca and in the rest of the distal intestine. Moreover, low proliferation corresponded to low frequency of apoptotic cells, while high proliferation corresponded to high apoptotic rate. These observations suggest that the proximal intestine and the apical part of the complex folds are subjected to a lower renewal rate compared to the pyloric caeca and the rest of the distal intestine. Apoptotic cells had two different morphologies: round and slender. Round cells correspond to the classical morphological features of apoptotic bodies already described in mouse small intestine [33]. They were localized along the villi length, at their tip as well as exfoliating into the intestinal lumen. Slender cells, with a strong eosinophilic cytoplasm, were observed only in the distal intestine, on the apical part of the complex folds and in the pyloric caeca. These features correspond to the early apoptotic stages in which dying cells assume the typical funnel-like structure with crescent nuclei and display a strongly acidophilic cytoplasm due to chromatin condensation [34].

## 5. Conclusions

Our data indicate that pyloric caeca, the villi and the basal portion of the complex folds in the distal intestine have a number of common morphological characteristics that set them apart from the proximal intestine and from the apical portion of the distal intestine complex folds. This indicates that rainbow trout intestine is characterized by the discontinuous distribution of two different morphological and functional compartments. Therefore, our results suggest that rainbow trout intestine is physiologically arranged to mingle digestive and absorptive functions along its length.

## Figures and Tables

**Figure 1 animals-10-00745-f001:**
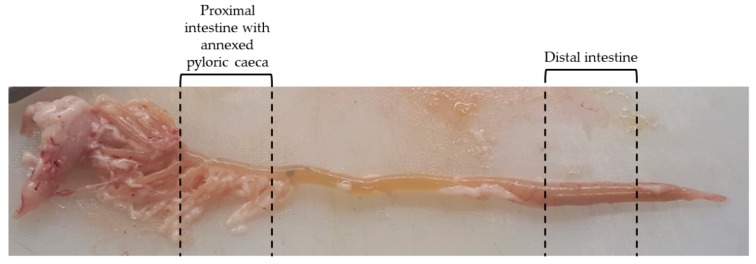
Selected regions of the samples collection.

**Figure 2 animals-10-00745-f002:**
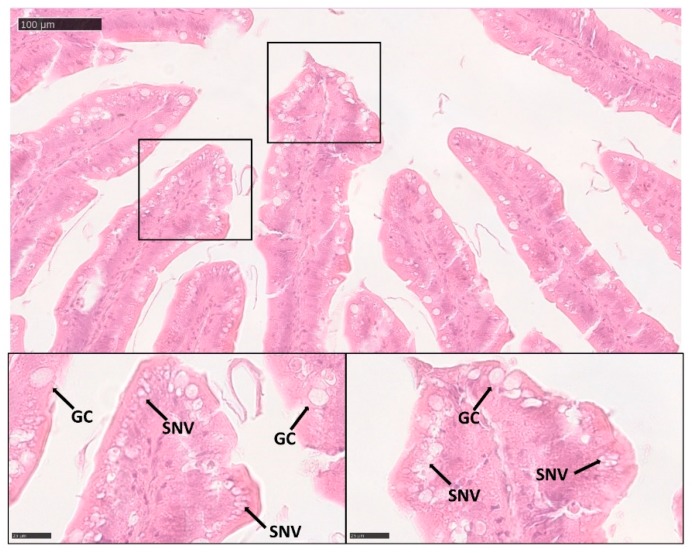
Hematoxylin/eosin (HE) stained section, showing the presence of enterocytes supranuclear vacuolization (SNV) and goblet cells (GC) in the pyloric caeca of 500 gr rainbow trout.

**Figure 3 animals-10-00745-f003:**
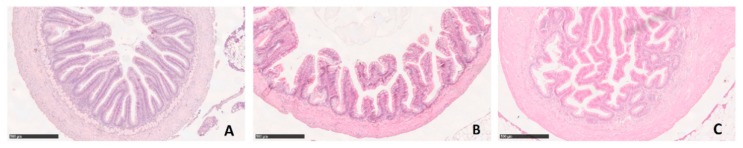
Branching of intestinal villi in the anterior intestine of rainbow trout during growth ((**A**) 50 g; (**B**) 150 g; (**C**) 500 g).

**Figure 4 animals-10-00745-f004:**
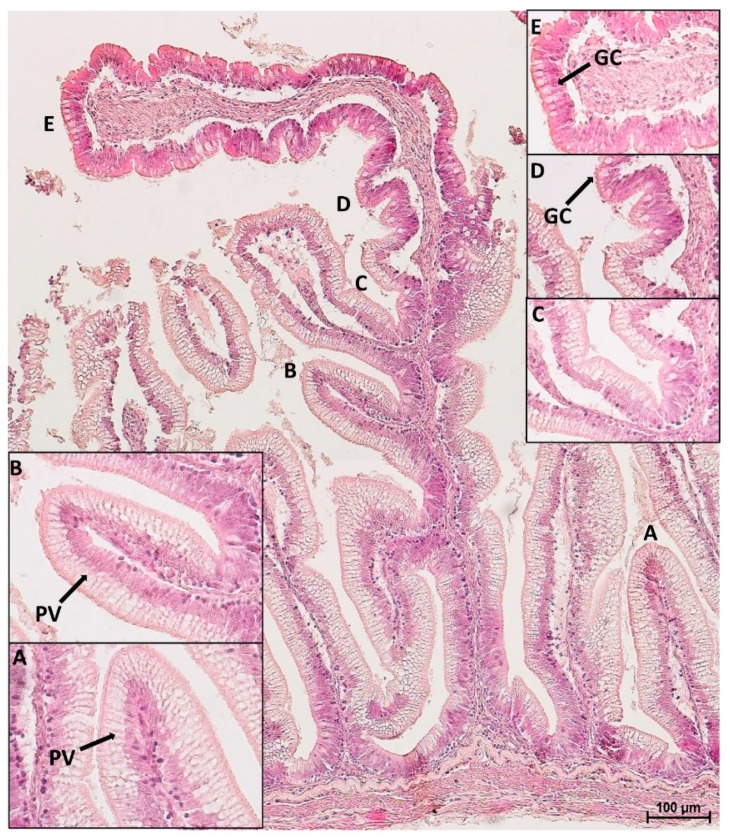
HE stained section of a complex fold in the distal intestine: parietal villi (**A**), as well as villi emerging from the basal part of the fold (**B**,**C**), were both covered by pinocytotic vacuoles (PV). At one point, the epithelium morphology along the fold, drastically changed: pinocytotic vacuoles tended to disappear (**D**). The apex of the fold presented goblet cells (GC) and non-vacuolated enterocytes (**E**).

**Figure 5 animals-10-00745-f005:**
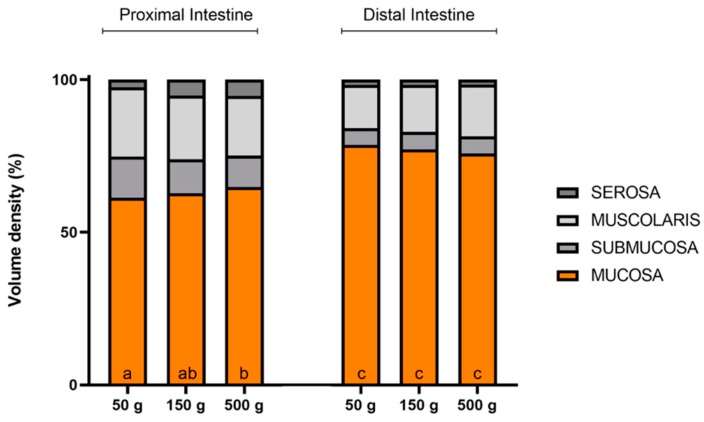
Volume density (Vv) estimation of rainbow trout’s intestinal layers in the proximal and distal intestine during growth. Data are expressed as a percentage of the whole intestinal wall. ^a–c^ Different superscripts in the same layer indicate significant differences (*p* < 0.05) determined by two-way ANOVA (animal weight and intestinal tract: independent variables).

**Figure 6 animals-10-00745-f006:**
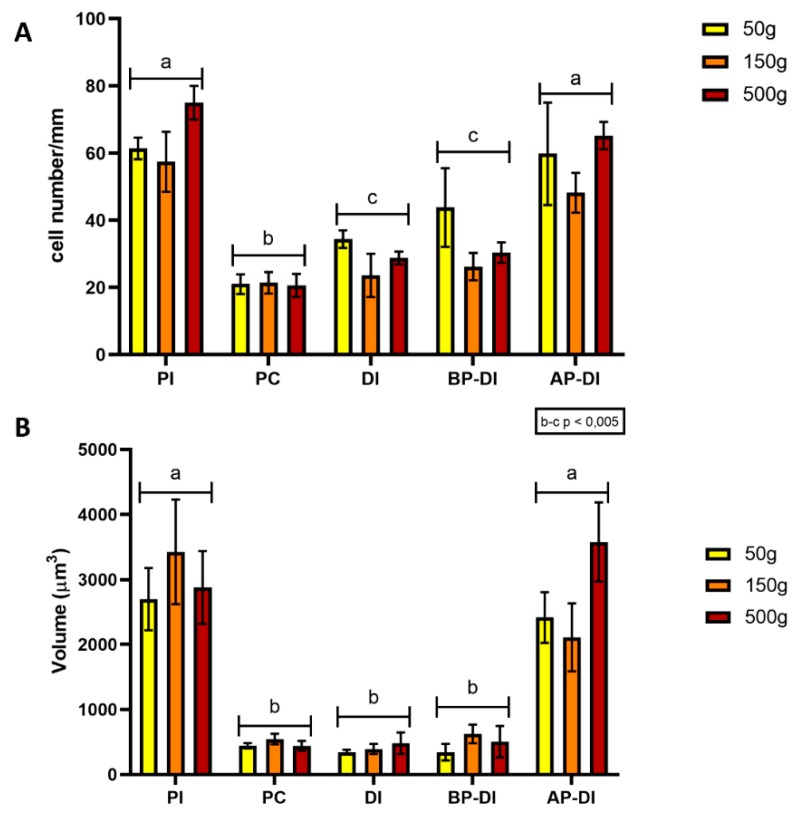
Goblet cells number (**A**) and volume (**B**) within five regions of the intestinal tract: proximal intestine (PI), pyloric caeca (PC), distal intestine (DI), basal (BP-DI) and apical part (AP-DI) of the complex folds of the distal intestine. Values are measured at three different stages of development (50, 150 and 500 g). ^a–c^ Different superscripts within the same histogram indicate significant differences (*p* < 0.05 or *p* < 0.005) measured by two-way ANOVA (animal weight and intestinal tract: independent variables).

**Figure 7 animals-10-00745-f007:**
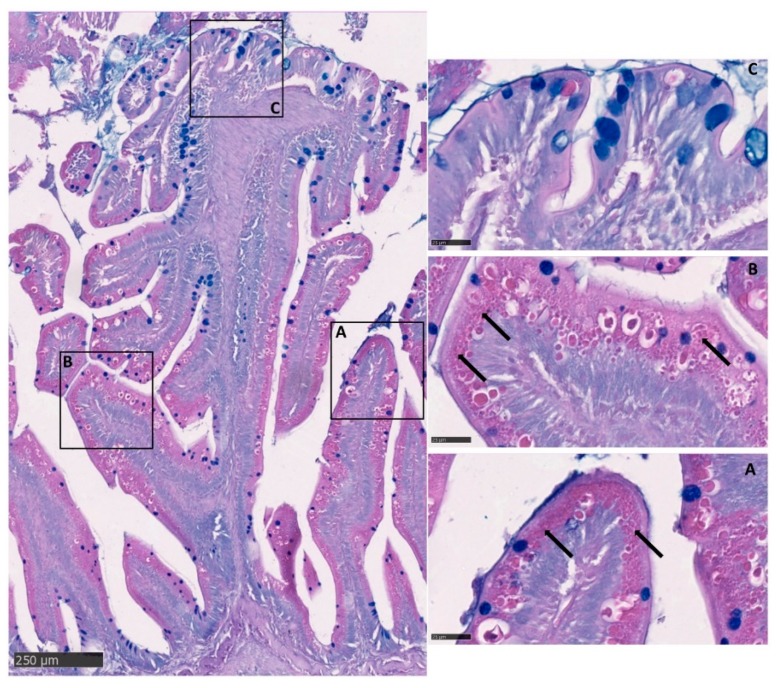
Alcian Blue–Periodic acid–Schiff (AB-PAS) stained section showing the presence of numerous PAS-positive pinocytotic vacuoles (arrows) on the villi of the distal intestine (**A**), as well as, on the basal part of the complex folds (**B**); whereas, they were completely absent at the fold apex (**C**).

**Figure 8 animals-10-00745-f008:**
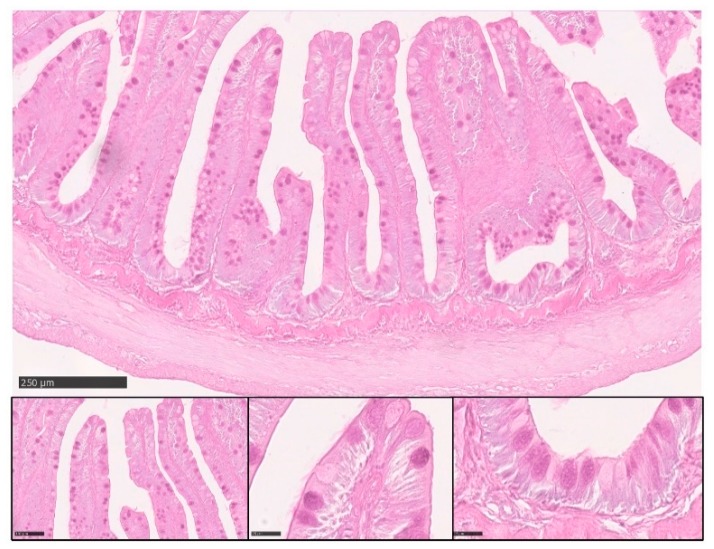
Representative figure of PAS stained section of the anterior intestine of rainbow trout.

**Figure 9 animals-10-00745-f009:**
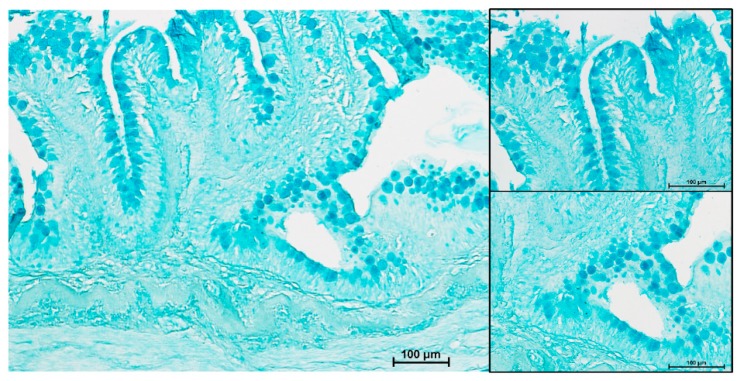
Representative figure of AB pH 2.5 stained section of the anterior intestine of rainbow trout.

**Figure 10 animals-10-00745-f010:**
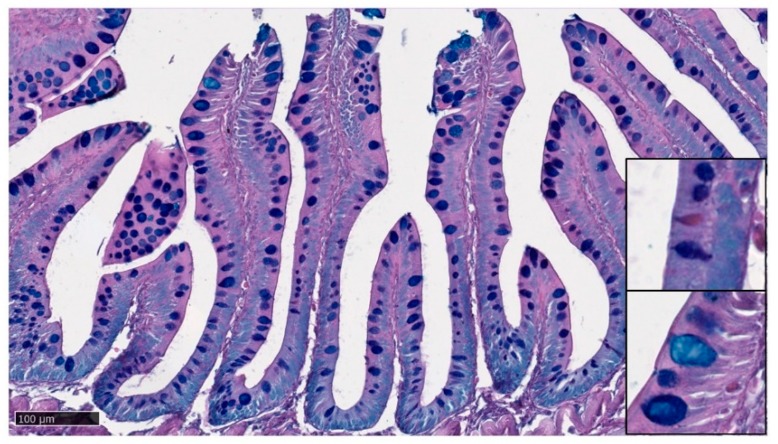
AB-PAS stained section showing the presence of a heterogenous populations of goblet cells. Most mucus-secreting cells produced a mixed secretion (PAS-AB positive-dark blue cells), some secreted acid mucins (AB positive-light blue cells), and a smaller amount produced neutral mucus (PAS positive-magenta cells).

**Figure 11 animals-10-00745-f011:**
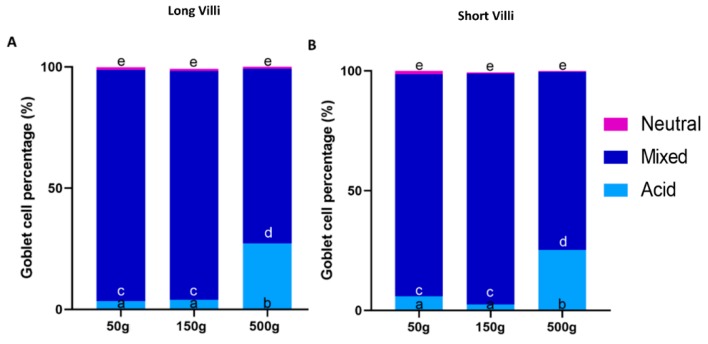
Estimation of the percentage of goblet cells secreting acid mucins (AB positive), mixed mucins (AB/PAS positive) and neutral mucins (PAS positive) in the proximal intestine of both short (**A**) and long (**B**) villi along rainbow trout development. ^a–e^ Different superscripts within the same histogram indicate significant differences (*p* < 0.05) measured by one-way ANOVA (animal weight: independent variable).

**Figure 12 animals-10-00745-f012:**
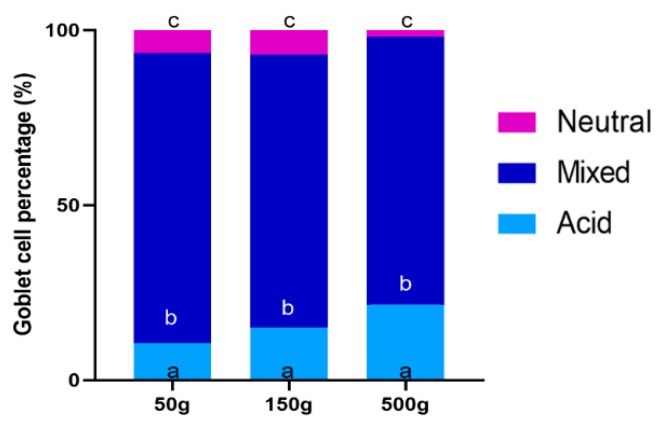
Estimation of the percentage of goblet cells secreting acid mucins (AB positive), mixed mucins (AB/PAS positive) and neutral mucins (PAS positive) in the pyloric caeca along rainbow trout development. In this region, the quality of mucus secretion remained constant along development. The same superscripts ^a–c^ within the same histogram indicate no significant differences (*p* > 0.05) measured by one-way ANOVA (animal weight: independent variable).

**Figure 13 animals-10-00745-f013:**
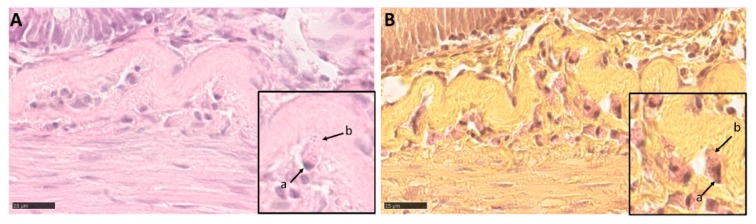
HE (**A**) and phloxine tartrazine (**B**) stained sections showing intestinal eosinophilic granule cells within the granulosum layers of submucosa (a—nucleus; b—eosinophilic granular cytoplasm).

**Figure 14 animals-10-00745-f014:**
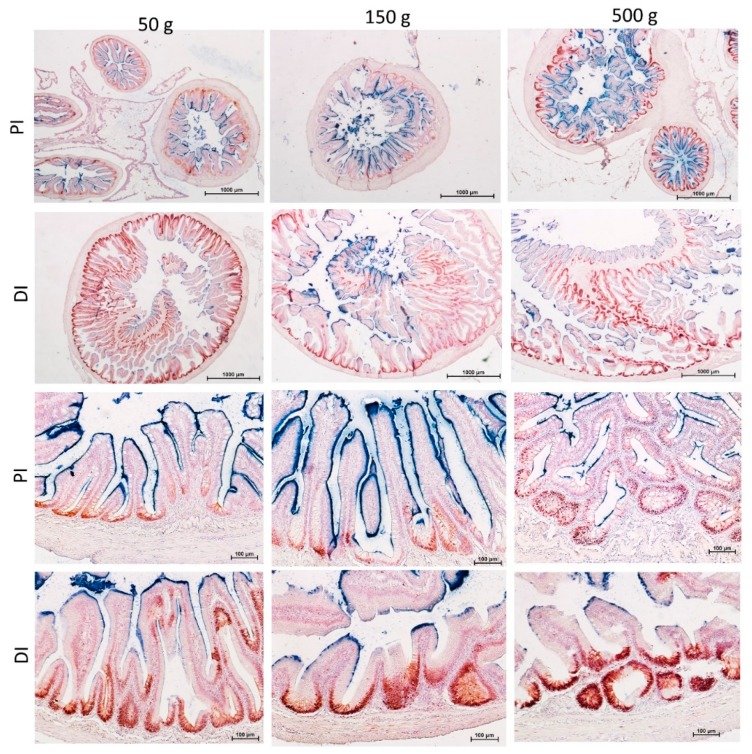
Proliferating cell nuclear antigen (PCNA) immunolocalization was observed within the intestinal folds at the basis of the villi in both proximal (PI) and distal (DI) intestine in all investigated stages. PCNA^+^ cells were also found in correspondence of villi branching points in the proximal intestine. Cell proliferation was more intense and widespread in the distal compared to the proximal intestine in all investigated stages. Fully differentiated enterocytes expressing alkaline phosphatase (AP) activity were observed at the villi apex, but conversely, its expression was stronger and more widespread in the proximal compared to the distal intestine in all ages.

**Figure 15 animals-10-00745-f015:**
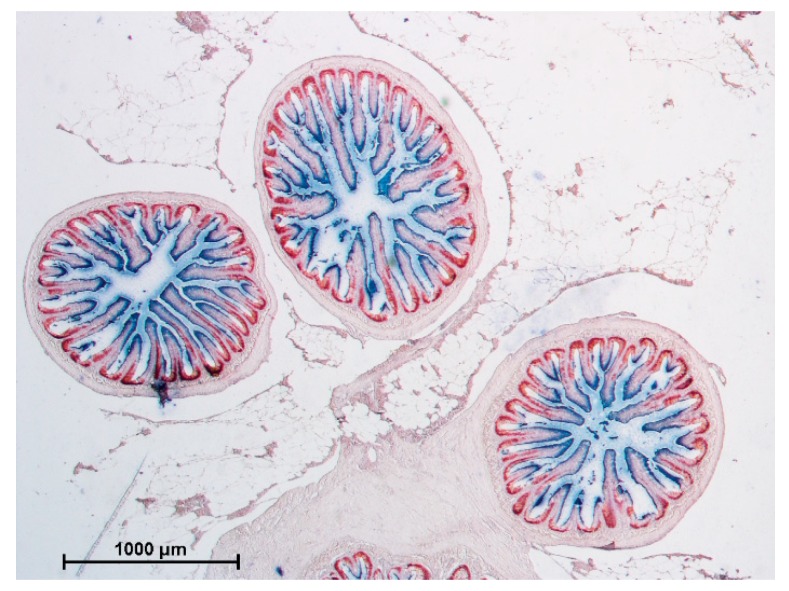
Representative figure of PCNA combined AP stained section pf pyloric caeca showing a proliferating pattern similar to that observed in the distal intestine and in the basal part of the complex folds.

**Figure 16 animals-10-00745-f016:**
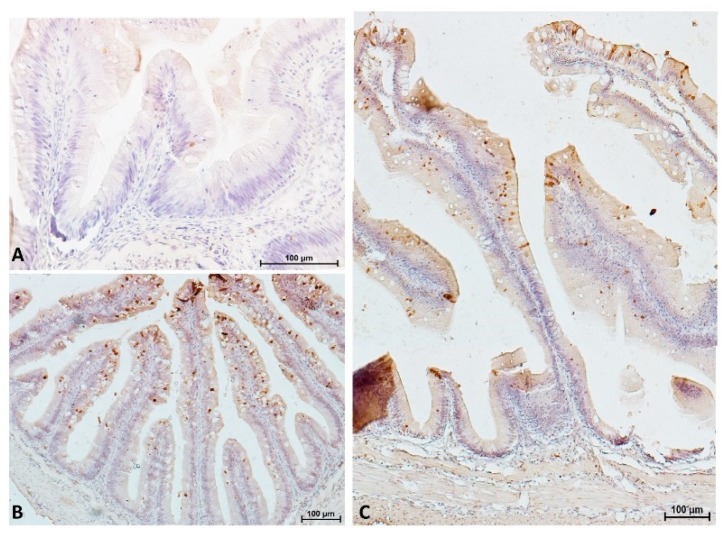
Representative figure of cells undergoing apoptosis showing only a few round apoptotic cells in the anterior tract (**A**) and many more apoptotic cells, both round and slender in the pyloric caeca (**B**) and in the posterior intestine (**C**).

**Figure 17 animals-10-00745-f017:**
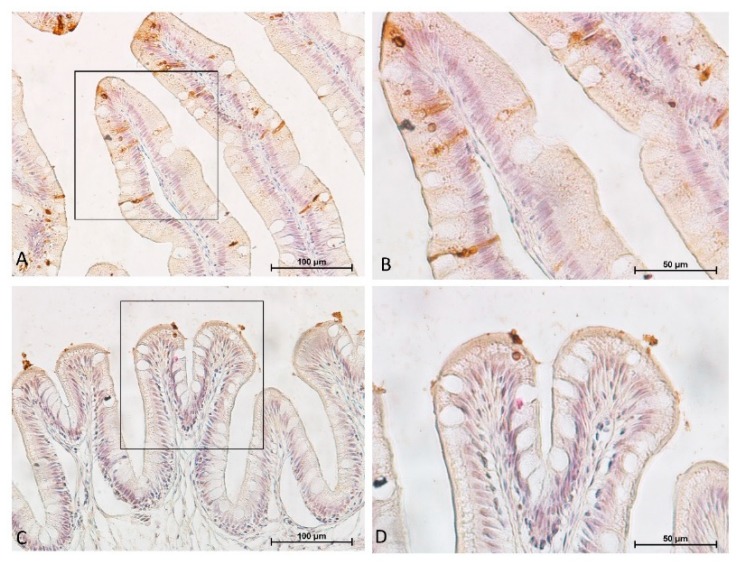
Representative figure of cells undergoing apoptosis showing numerous, both round and slender apoptotic cells in the basal part of the complex folds (**A**,**B**) and only a few round apoptotic cells in the apical part of the complex folds (**C**,**D**).

**Figure 18 animals-10-00745-f018:**
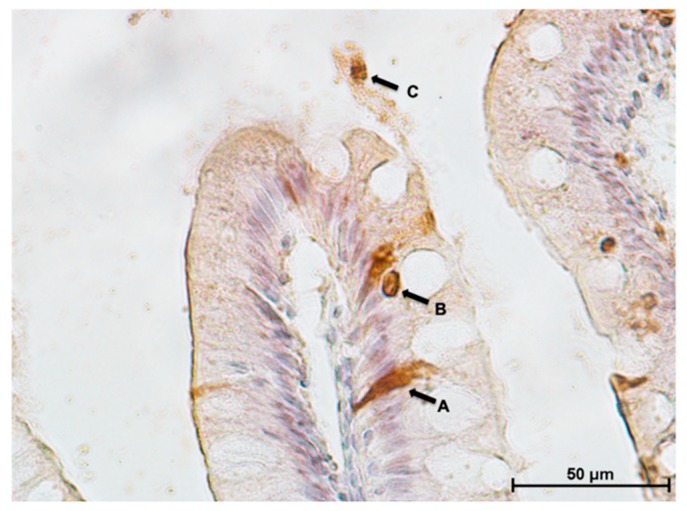
In situ detection of cells undergoing apoptosis showing three different apoptotic stages: A: early apoptotic stage; B: apoptotic body: C: cells exfoliating into the intestinal lumen.

**Figure 19 animals-10-00745-f019:**
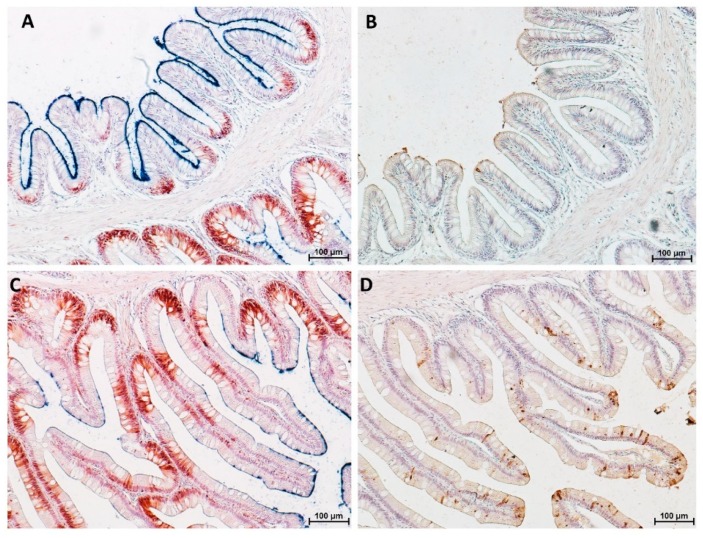
Representative figure of PCNA immunolocalization (red), histochemical detection of differentiated cells (blue) and in situ detection of apoptotic cells (brown). The apical part of the complex folds was characterized by a low PCNA expression, few round apoptotic cells and a strong alkaline phosphatase signal (**A**,**B**). In contrast, high proliferation and high apoptotic rate associated with low alkaline phosphatase expression in the basal part of the complex folds were observed (**C**,**D**).

**Table 1 animals-10-00745-t001:** Gross anatomy measurements in rainbow trout along the first year of development.

Trout Weight (g)	N.	Rainbow Trout Length (cm)	Whole Intestinal Length (cm)	Intestinal Segment Length (cm)	
				Proximal	Distal
50	5	16.8 ± 0.29	9.7 ± 0.62	6.0 ± 0.30	3.6 ± 0.28
150	5	22.7 ± 0.58	15.1 ± 0.80	9.0 ± 0.30	6.0 ± 0.50
500	5	32.4 ± 0.53	26.1 ± 0.76	18.2 ± 0.64	7.9 ± 0.11

Values are expressed as mean ± SD.

**Table 2 animals-10-00745-t002:** Evaluation of pyloric caeca histometry in rainbow trout along the first year of development.

Trout Weight	N.	Pyloric Caeca (µm)	Enterocytes Supranuclear Vacuolization (SNV)
50 g	5	380.91 ^a^ ± 35.85	−
150 g	5	719.50 ^b^ ± 32.45	−
500 g	5	1011.50 ^c^ ± 18.14	+

Values are expressed as mean ± SD. ^a–c^ Different superscripts in the same column indicate significant differences (*p* < 0.05) determined by one-way ANOVA (animal weight independent variable). The presence or the absence of enterocytes supranuclear vacuolization are indicated with “+” or “−” respectively.

**Table 3 animals-10-00745-t003:** Evaluation of proximal intestine histometry in rainbow trout along the first year of development.

Trout Weight (g)	Proximal Intestine				
	N.	Short Villi Length (µm)	Long Villi Length (µm)	Villi Width (µm)	Villi Branching
50	5	251.0 ^a^ ± 13.5	470.4 ^a^ ± 64.6	96.0 ^a^ ± 11.4	−
150	5	310.3 ^a^ ± 44.5	555.8 ^a^ ± 64.2	122.9 ^b^ ± 10.1	+
500	5	250.3 ^a^ ± 42.1	657.8 ^b^ ± 49.1	116.9 ^b^ ± 14,4	++

Values are expressed as mean ± SD. ^a–b^ Different superscripts in the same column indicate significant differences (*p* < 0.05) determined by one-way ANOVA (animal weight independent variable). The absence or the occurrence of villi branching is indicated with “−” and “+” respectively. “++” indicate an increase in villus branching.

**Table 4 animals-10-00745-t004:** Evaluation of distal intestine histometry in rainbow trout along the first year of development.

			Distal Intestine	
Trout Weight (g)	N.	Short Villi Length (µm)	Long Villi Length (µm)	Folds Height (µm)
50	5	−	−	−
150	5	211.1 ± 3.1	592.4 ± 19.3	1284.9 ± 51.0
500	5	202.4 ± 20.1	556.5 ± 37.0	1410.1 ± 201.7

Values within the same column indicate no significant differences (*p* > 0.05) determined by one-way ANOVA (animal weight independent variable).

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
