# Peer review of "A Detailed Study of Rainbow Trout (Onchorhynchus mykiss) Intestine Revealed That Digestive and Absorptive Functions Are Not Linearly Distributed along Its Length"

_animals, 2020, doi:10.3390/ani10040745_

Round 1

Reviewer 1 Report

In the present form the maniscript is accetable for publication

Reviewer 2 Report

No specific comment or suggestion to the authors. The paper submitted in the revised version satisfy all the main reviewer requirements and is suitable for publication.

This manuscript is a resubmission of an earlier submission. The following is a list of the peer review reports and author responses from that submission.

Round 1

Reviewer 1 Report

Comments for animals-736618 Manuscript

Title: A detailed study of rainbow trout (Oncorhynchus mykiss) intestine reveals the existence of two morphological and functional districts in its distal portion. This study was conducted to study characterized in the detail the intestinal epithelium to establish accurate reference values that could increase the sensitivity of current gut health endpoints in response to diet modifications. The results might provide some help to the sustainability of the trout farming. Here, some comments needed to be resolved as follows.

1. The title

The title should be revised to show the main content of this manuscript.

2. Abstract

The abstract also should be revised. Change the main description of the Materials and Methods to be the main results they found. And then, giving a conclusion at the end of the Abstract.

3. Introduction

Change the one paragraph to two paragraphs.

Reviewer 2 Report

Manuscript Title

A detailed study of rainbow trout (Oncorhynchus mykiss) intestine reveals the existence of two morphological and functional districts in its distal portion.

Authors performed the characterization of the epithelial cells lining the intestinal mucosa in rainbow trout along the first year of development. They also studied the absorptive and secretory activity of the intestinal mucosa and its ability to renewal. Moreover, authors described the presence of two functional compartments within the distal intestine.

Authors report: “Samples of proximal and distal intestine were collected from five female rainbow trout (Oncorhynchus mykiss) with different ages, (7,10 and 12 months) weighing approximately 50, 150 and 500 g, respectively”.

In this sentence it is not clear how many animals have been used for each age. Furthermore, there is no information on breeding conditions, for example the water temperature. It is somewhat difficult to think that the animals tripled their weight in two months (from 10 to 12 months).

“Several unopened pieces were taken from both proximal and distal intestine.”

There is no information regarding sampling, the starting and ending point of each intestinal part, the relative length of each segment at 7, 10, 12 months. All the staining performed in this study should be carried out in pyloric caeca too. Authors report that the mucosa volume was significantly higher in the distal than in the proximal part of the intestine at all ages and remained unchanged during development. It is therefore not clear why it was impossible to collect samples of distal intestine from individuals of 50 grams. In the materials and methods it is not reported which antibody was used for immunohistochemical analysis (anti-….., monoclonal, polyclonal, species, dilution). Finally, there is no a positive control. The results of the histochemical analysis are limited. Glycoconjugates characterization needs to be performed with the following conventional histochemical stains: PAS, AB (pH, 2.5), AB-PAS, AB (pH, 1), AB (pH, 0.5). There are two figures 1, two figures 2 and 2 figures 3 The magnification of the figures concerning apoptosis and immunohistochemistry should be increased.

Reviewer 3 Report

The study  entitled “A detailed study of rainbow trout (Oncorhynchus 2 mykiss) intestine reveals the existence of two 3 morphological …. “ by Vendile et al. is an interesting, well written anatomical investigation of fish gut. The manuscript needs only minor revisions regarding result section.

Please, change the title of paragraph 3.2 in “Microscopical anatomy”

Figure 3 must replace Figure 1 (which is repetitive of figure 3).

Line 203. Does the phrase “No supranuclear vacuoles were observed….” Refer to proximal intestine? In the rest of the text is assumed the contrary. Please, check this point.

Line 214: delete the phrase “Morphometric …..500 g individual”

Line 236. Explain “compactum layer” (dense connective tissue?) and “granulosum layer” (it is layer of epidermis…)

Line 241 delete fiber and change in “arranged muscle cells”

Figure 9 The image is out of focus and to visualize EGCs high magnifications are mandatory

Line 337: Numeration of the figures after this line is wrong: check carefully

Reviewer 4 Report

Manuscript title

A detailed study of rainbow trout (Oncorhyncus mykiss) intestine reveals the existence of two morphological and functional districts in its distal portion.

Authors characterized the intestinal mucosa epithelial lining in rainbow trout during the first year of life. The authors emphasized the existence of two morphs - functional compartments in  the distal intestine. The absorptive and secretory ability and the renewal capacity of the intestinal mucosa was also investigated. Unfortunately there is not any information to identify starting and ending point of distal intestine. For that concern glycoconjugates characterization I should like some separate pictures of PAS, Alcian B -PAS, Alcian B at different pH (0,5 -1,0 - 2,5).

Finally, statistical analysis section lack some important information. 

Authors should indicate whether assumption for ANOVA were satisfied and how did they check their validity. Authors should specify the dependent variables tested and the respective independent categorical variables used in the two-way ANOVAs. Without this information is impossible understand if the analyses were performed correctly.

Results are not enough clearly presented. In Table 1 and 2 and in Fig.5 Authors do not indicate the meaning of the letters that indicate statistical significant differences. In the captions of Tables 1 and 2, in Figs. 5, 6 and 8 authors should indicate what statistical test applied.
